# A Cross Sectional Study to Identify Traumatic Stress, Medical Phobia and Non-Adherence to Medical Care among Very Young Pediatric Patients

**DOI:** 10.3390/ijerph20021122

**Published:** 2023-01-08

**Authors:** Amichai Ben-Ari, Yaron Sela, Shiri Ben-David, Yael L. E. Ankri, Fortu Benarroch, Roy Aloni

**Affiliations:** 1Department of Behavioral Sciences, Ariel University, Ariel 40700, Israel; 2Herman Dana Division of Child and Adolescent Psychiatry, Hadassah-Hebrew University Medical Center, Jerusalem 91240, Israel; 3School of Psychological Sciences, Tel Aviv University, Tel Aviv 69978, Israel; 4Department of Psychology, The Hebrew University of Jerusalem, Jerusalem 91240, Israel; 5Department of Psychiatry, Hadassah Hebrew University Medical Center, Jerusalem 91240, Israel

**Keywords:** medical phobia, traumatic medical event, pediatric post-traumatic stress disorder

## Abstract

After a traumatic medical event, such as surgery or hospitalization, a child may develop a phobia of medical care, sometimes preventing future medical adherence and impairing recovery. This study examined the correlation of Pediatric Medical Traumatic Stress (PMTS) on the development of Medical Phobia (MP) and subsequent treatment adherence. We enrolled 152 parents of children aged 1–6 hospitalized in a surgical ward. During hospitalization, parents completed questionnaires that identified post-traumatic stress symptoms. Four months post hospitalization, parents completed questionnaires on post-traumatic stress, medical phobia, psychosocial variables and medical adherence. We found a positive correlation between PMTS and MP and low adherence to medical treatment. In addition, MP mediated the relationship between PMTS severity and adherence, indicating that PMTS severity is associated with stronger medical phobia, and lower pediatric adherence to medical treatment. Our findings suggest that medical phobia serves as an essential component of PMTS. It is important to add medical phobia to medical stress syndrome definition. In addition, as MP and PMTS are involved in the rehabilitation and recovery process and subsequent success, it is an important aspect of treatment adherence.

## 1. Introduction

While most children undergoing a serious medical incident will gradually heal with no lingering psychological symptoms, 16–28% will develop post-traumatic stress symptoms (PTSS), which may influence both their physical recovery and future functioning [1,2]. Very young children [3] have different types of pain expression than adults and even older children [4] and may be especially vulnerable to PTSS, which has both behavioral [3,5] and physical [6] development consequences.

Although not yet defined as a separate category of trauma, as are pediatric PTSS and PTSD, Pediatric Medical Traumatic Stress (PMTS) has been identified as a post-traumatic response of a child to a medical procedure, incident or illness [2,7,8,9,10]. As standardized ways to measure PMTS have not yet been adopted [11], these responses are often defined in terms of PTSS or post-traumatic stress disorder (PTSD) diagnoses [2], yet PMTS is becoming recognized as a specific type of PTSS by clinicians who work with children after a serious medical event [2,7,9,12,13,14].

Children who suffer from PMTS may also engage in avoidance, as part of the stress response together with re-experiencing the trauma and hyperarousal [4,15]. Avoidance is highly correlated with lack of medical adherence, which may compromise taking medications, attending clinic visits, preventing infections and reducing or preventing side effects [16]. Medical nonadherence is a significant problem among children, with some meta-analyses suggesting up to 50% nonadherence among children with chronic illness [17]. Even young children, ages 1–6, have been identified as being nonadherent, specifically in the case of asthma [18] and other chronic illness [19]. There is evidence to support many reasons for medical nonadherence in children, including psychosocial factors and lower family cohesion [20]. However, presence of PTSS symptoms, also categorized as PMTS, has been identified among children as a possible predictor of nonadherence among pediatric transplant patients [20,21] and children with chronic physical illness [22].

Medical phobia may be another clinical manifestation of the traumatic medical event, appearing with great frequency with PMTS [23]. In fact, medical phobia may be a factor that explains the connection between a traumatic medical event and medical adherence [24]. Phobia is defined as intense and irrational anxiety or fear of a specific object or situation that evokes in the individual a powerful and disproportionate fear [25]. Medical phobia is defined as fear of exposure to a medical procedure (e.g., injection [26], medical examination, dental treatment [27], or fear of injury [25] or even fever in a child [28]. 

Despite the growing body of research on PMTS and medical nonadherence among children, less is known about very young children and this association, especially after a medical intervention such as surgery. Additionally, little is known about the role that medical phobia may play in the interaction.

The purpose of the study was to examine the consequences of traumatic medical events on the development of medical phobia in children. We hypothesized that PMTS symptoms would correlate with increased presence of medical phobia and lower adherence to medical treatment. We also hypothesized that this phobia would act as a mediating factor of PMTS and adherence to medical treatment.

## 2. Materials and Methods

### 2.1. Participants

Parents of very young children (ages 1–6) who were hospitalized in a post-surgical ward were approached to participate in this study. In order to ensure that the research sample was representative of the total ward population, we compared socio-demographic and hospitalization data of the sample group with those of the overall ward population (ages 1 to 6; N = 6231) from the previous year (Table 1). 

### 2.2. Measures

Socio-Demographic Measurement: This questionnaire included age, child gender, gender of the participating parent and socio-economic status. Socio-economic status was determined by weighting the parents’ answers to the following questions: years of parental education, number of rooms in the house, and presence of any financial difficulties (self report).

The Child Medical Fear Scale (CMFS) [29]: This questionnaire measures the child’s response to medical events experienced as stressful and frightening. The questionnaire has 17 items, and for each item there were three options: 0 = not afraid at all, 1 = a little scared, and 2 = very scared. Translation validity was conducted by back translation and expert review. We found good internal reliability at both phases of the study [T1 = (0.82) and T2 = (0.78)].

Pittsburgh Rehabilitation Participation Scale [30]: A questionnaire for measuring adherence and non-adherence to rehabilitative treatment, it measures the child’s level of motivation according to a 5-point Likert scale and also addresses the effect of organic and psychological measures on the level of responsiveness. This questionnaire has two items and was completed by the child’s parents in this study. Translation validity was conducted by back translation and expert review. It was found to have a high reliability of 0.91 and a prediction validity of r = 0.32, *p* < 0.001.

UCLA PTSD Reaction Index for DSM-5 Parent/Caregiver Version for Children Age 6 Years and Younger [31]: This measure possesses both high validity and reliability to assess PTSD among a variety of ethnic and religious groups. It consists of 22 items. In this study, the internal reliability scale (Cronbach’s α) was α = 0.95.

Young Child PTSD Checklist (YCPC) [3]: This assessment measures PTSD as defined by the DSM-5 among very young children and was completed by the parent or caregiver on behalf of the child. This questionnaire contains 42 items; 13 questions on type of pediatric traumatic experience; 23 questions on pediatric symptom experience; and 6 items regarding functional impairment. We identified good internal reliability for symptom scales (Cronbach’s α): arousal: α = 0.92, avoidance: α = 0.93, reliving: α = 0.88, total score: α = 0.97. 

Psychosocial Assessment Tool II (PAT 2.0) [32]: This measure is designed to assess psychosocial problems in children with life-threatening diseases. The instrument includes 84 items and comprises 7 sub-scales: Family Structure and Resources, Family Social Support, Family Problems, Parent Stress Reaction, Family Beliefs, Child Problems and Sibling Problems. Each item in the sub-scales is dichotomously scored. Higher scores are associated with higher levels of psychosocial risk for the child. For this analysis, we utilized the subscales of Child Problems and Parent Stress Reaction. 

### 2.3. Procedure

After receiving ethics approval from the sponsoring organization, during consecutive hospitalizations of 204 children from April to July 2018, 184 parents agreed to participate in the study. An explanation of the study was provided, and parents signed a consent form and completed the questionnaires with the help of researchers during a face-to-face interview. Six parents refused to participate because of the emotional costs; 28 parents cited lack of time for their refusal.

Stage (1) 152 parents completed the UCLA [31] and YCPC [3] questionnaires (identifying presence and level of PMTS). Both questionnaires were used for analysis in order to achieve convergent validity. As our study population was very young, the parents completed all the questionnaires based on their children’s responses and experiences. 

Stage (2) Four months after discharge, 152 parents completed the UCLA [28] and YCPC [3] questionnaires (measuring PTMS), the CMFS [29] (level of fear/medical phobia), the PAT 2.0 (to measure overall child problems and caregiver stress) [32] and the Pittsburgh Questionnaire [30] to assess pediatric adherence to medical care.

### 2.4. Data Analysis

Data were analyzed using SPSS v. 27 (IBM, Chicago, IL, USA). Descriptive statistics were produced using frequencies for categorical variables and means with standard deviations for continuous variables. First, we assessed associations between variables by using Pearson correlations. Second, we performed hierarchical multivariate regressions that identified correlations among trauma severity, medical phobia and adherence to medical treatment. We controlled for multiple comparisons and used variable scores generated from Stage 2 of the study only. Finally, to assess the mediation model, the Structural Equation Modeling (SEM) was implemented. The following indices were used to evaluate the model: chi-squared, which is acceptable when the value is not significant; the goodness of fit index (GFI), the comparative fit index (CFI), and the non-normed fit index (NNFI), (adequate values—above 0.90, excellent fit—above 0.95); and the root mean square error of approximation (RMSEA) (adequate values—less than 0.08, excellent fit—less than 0.06) [33]. Level of significance (*p*-value) was 5%.

## 3. Results

For this analysis, we included 152 parent and child dyads (one parent completing the questionnaire on behalf of themselves and their child) who completed the questionnaires at both stages. This sample included 58 girls (38.2%), 94 boys (61.8%), 61 (41%) fathers and 91 (59%) mothers. The children were between one and six years old (M = 2.89, SD = 1.51). One hundred and three children had been hospitalized for elective surgery (68.2%), and the rest for emergency interventions (*n* = 48, 31.8%). Hospitalization time ranged from 1–37 days (M = 4.71, SD = 5.70).

As hypothesized, PTMS symptoms correlated with higher medical phobia scores (r = 0.23, *p* < 0.01) and lower adherence to medical treatment (r = −0.46, *p* < 0.01). A moderate negative correlation was found between the child’s medical phobia score and adherence to medical treatment (r = −0.22, *p* < 0.01) (Table 2).

As for the second hypothesis regarding medical phobia as a mediating factor of PMTS and adherence to medical treatment, Table 3 presents the results of hierarchical multivariate regressions that show correlation among PTMS symptoms, the presence of medical phobia and adherence to medical treatment. The presented hierarchical regressions aimed to provide information regarding the unique contribution of three sets of variables to the explanation of outcomes: (1) demographics (2) surgery-related variables and (3) psychological variables. Variables were entered in three steps. In step 1, we entered the child’s gender and age; in step 2, we entered clinical and hospitalization variables. In step 3, we entered the child’s problems score, parental stress reaction score, the UCLA scores that measured PMTS and medical phobia scores (in separate regressions).

The results showed that age and gender were not correlated with PMTS symptoms. However, when entering clinical and hospitalization variables, it was found that hospitalization duration (>4 days) (β = 0.54, *p* < 0.05), low socio-economic status (β = 0.26, *p* < 0.05) and previous painful procedures (β = 0.32, *p* < 0.05) were positively correlated with PMTS. Finally, as found in step 3 of regression, child’s problems score was positively related with PMTS symptom severity (β = 0.24, *p* < 0.01) but parental stress reaction was not. As for medical phobia, the results showed that after adjusting for demographic and hospitalization variables, PMTS was positively related with presence of medical phobia (β = 0.43, *p* < 0.01). Adherence to medical treatment was negatively associated with PMTS (β = −0.44, *p* < 0.05), and positively associated with medical phobia (β = 0.22, *p* < 0.05) (Table 3).

To assess how medical phobia mediates the relationship between pediatric medical traumatic stress syndrome (PMTS) and adherence to medical treatment, we conducted a path analysis by Structural Equation Modelling (SEM). The model yielded fair goodness of fit indices (χ^2^(1) = 65.61; *p* < 0.001; GFI = 0.88; NFI = 0.87; CFI = 0.88; RMSEA = 0.09). Specifically, PMTS was associated with a stronger presence of medical phobia (β = 0.14, *p* < 0.05), which, in turn, was negatively correlated with the child’s reduced adherence to medical treatment (β = −0.22, *p* < 0.01). A partial mediation effect was found, as well as direct (β = −0.05, *p* < 0.05) and indirect effects (β = −0.46, *p* < 0.05).

## 4. Discussion

Our study provides important evidence about the correlation of pediatric PMTS, and treatment nonadherence among very young children, and the possible mediating role that medical phobia may have in this correlation. Demonstrating how traumatic stress from a medical event, and the presence of a medial interact that could impair pediatric medical adherence, is an important addition to the body of literature regarding mental health effects of a serious medical event on children.

Our findings are consistent with other studies that have shown a correlation between development of a phobia and a traumatic event in children [34,35] and PMTS and medical avoidance [15,36], which is a precursor to medical phobia. Our results are also in agreement with studies that found a correlation between PTSS or PTSD symptoms and lack of medical adherence among adults [37]. 

Several analyses about dental phobia, specifically, have suggested that in both adults and children, this specific anxiety stems not as an irrational fear, but rather as a response to a traumatic encounter with a dentist, which included feelings of fear, pain and lack of control as an expression of traumatic distress [38]. Perhaps this perspective can be applied to medical phobias in general. A broad and accurate view suggests treating medical phobia as another component of pediatric medical stress syndrome. From this position, one should consider adding a reference to medical phobia as an essential component in the definition of PMTS. It is possible that this understanding may serve as a basis for justifying the use of a PMTS diagnosis for children who have developed post-traumatic symptoms after a medical procedure and do not meet all the criteria for a PTSD diagnosis. The fact that medical phobia is a key feature in the diagnosis of PMTS can reinforce the decision to add it as a new diagnosis. This contrasts with the current suggestion of using partial diagnoses, such as post-traumatic stress symptoms (PTSS) or partial PTSD, when discussing PMTS.

It may also be advisable to create an educational program for medical staff about the effects of surgical interventions and hospitalization on a child’s mental state, possibly resulting in future avoidance of medical care. In this context, it is necessary to provide staff with techniques on how to increase adherence among children, who may be at risk for developing PMTS and subsequent medical phobia.

Traumatic medical experiences and the consequent development of medical phobia may lead to the development of avoidance patterns and the child’s refusal to cooperate with future treatments and examinations. This has implications, both for the individual patient and for the system in general. Specifically, the child’s refusal to cooperate in future treatments and medical examinations could impair the child’s rehabilitative process, with negative prognostic consequences. This impact has been documented among pediatric oncology patients who have reduced adherence to important chemotherapy medications [36], as well as young children with asthma [18] and other chronic illness [19]. Systemically, lack of adherence affects the entire health system, adding high financial costs to the system [39]. Therefore, there is a need to minimize possible pediatric traumatic experience during hospitalization and swiftly identify possible sequelae of PMTS and medical phobia. 

The diagnosis of PMTS refers to the symptoms of post-traumatic stress disorder that developed following an illness or medical hospitalization. The findings of the current study show that the development of medical phobia and the subsequent impairment in compliance with reaching treatment are part of the consequences of such an event. Because of this, we propose to include the components of medical phobia and avoidance of treatment as part of the definition of PMTS. These components are unique to PMTS and different from what happens in other cases of PTSD, so it is important to include them when diagnosing PMTS [11].

### Limitations

Our study took place in only one hospital ward. Further research should focus on medical centers and additional populations in order to expand the external validity of the findings. However, the sample in the study included a wide range of children undergoing different types of surgical hospitalizations. Moreover, the sample included a wide range of population characteristics (socioeconomically), which does increase the applicability to other populations. Additionally, we assessed adherence to medical care as the care parents provided the child at home. This type of care is fundamentally different than assessment of adherence to hospital care. When a child receives home treatment, there is moderation of medical phobia that may reduce the avoidance. As we examined the behaviors and experiences of preschool-aged children, we also had the possible bias of parental reporting on their child’s thoughts and feelings, which may not accurately reflect the child’s actual perceptions. Additionally, there is an inherent bias with a single self report of all outcome variables rather than multiple informants/objective report methods [19]. 

## 5. Conclusions

In order to reduce the possibility of developing medical phobia, a follow-up study should be conducted to examine factors that explain the variability in presence and severity of medical phobia, interference with adherence to future hospital care, and possible preventive care. Additionally, medical phobia has a wide range of symptoms and manifests in many ways, including fear of injury, blood, doctors, hospitals and injections. Some studies have addressed only some of those expressions [26,40]. In terms of adherence, our study was only able to focus on medication taking. Because our population centered on preschool-aged children, we recognize that the degree of the child’s medical adherence is strongly related to the parents. However, when a child throws a tantrum before a medical examination, or refuses to take medication, some parents may not know how to cope with these refusals, and therefore the child will not adhere to the treatment regime [36,37]. Specifically for these parents, behavioral modification interventions may be helpful, but further investigation is needed to determine specific barriers to adherence among this subset of preschool children [41].

The findings of the present study indicate that medical phobia is positively and significantly related to PMTS and that it is a mediating factor between medical hospitalization and compliance with treatment. This study is an important addition to the research literature due to the fact that it specifically examines the consequences of traumatic distress on the patient’s level of compliance to attend follow-up treatments and cooperate with the treatment plan. Since it is known that a lack of cooperation is a significant barrier to recovery, there is significant value in the findings of the current study that shed light on the understanding of the dynamics that lead to this pattern. We recommend that further research be conducted regarding other aspects of medical phobia among preschool-aged children, specifically, and, subsequently, which aspects may be correlated to both post medical stress syndrome and lack of treatment adherence.

## Figures and Tables

**Table 1 ijerph-20-01122-t001:** Demographic and clinical background characteristics of the sample.

Variable	Mean	SD	N	%
Gender (Males)			94	61.8
Age	2.89	1.52		
Surgery complexity				
Low			91	64.1
Moderate			46	32.4
High			5	3.5
Hospitalization duration	2.60	2.00		
Socio-Economic status				
Low			34	22.2
Moderate			93	61.5
High			25	16.3
Child’s problems	1.21	0.25		
Parental stress reaction	0.47	0.58		
Medical operation avoided by child				
Ingestion of drops			23	15.0
Ingestion of medication			34	22.2
Bandages			32	20.9
Infusion and injections			10	6.5
Ointments			6	3.9
Nutrition/food restriction			10	6.5
Physiotherapy			7	4.6

**Table 2 ijerph-20-01122-t002:** Means, standard deviations and Pearson correlations between main study variables.

	M	SD	1	2	3
1. UCLA PTSS severity	7.82	10.87			
2. YCPC PTSS severity	7.73	9.69	0.90 **		
3. Child’s medical phobia	1.60	0.60	0.23 **	0.24 **	
4. Child’s adherence to medical treatment	3.76	1.89	−0.46 **	−0.49 **	−0.22 **

** *p* < 0.01.

**Table 3 ijerph-20-01122-t003:** Standardized coefficients of linear regression showing correlation of PTSS (UCLA), medical phobia and adherence to medical treatment.

		PTSS Severity (UCLA)			Medical Phobia			Adherence to Medical Treatment		
Step		Model 1	Model 2	Model 3	Model 1	Model 2	Model 3	Model 1	Model 2	Model 3
1	Gender (Males)	0.13	0.04	0.05	−0.11	−0.18	−0.20	−0.25	−0.25	−0.22
	Age	−0.20	−0.16	−0.24	−0.14	−0.09	−0.02	0.12	0.08	0.03
2	Surgery complexity		0.06	0.11		0.20	0.16		−0.11	−0.12
	Hospitalization duration		0.54 **	0.49 **		0.18	−0.08		0.04	0.27
	Low Socio-Economic status		0.26 *	0.22 *		0.13	−0.01		0.06	0.17
	Previous painful medical procedures		−0.32 *	−0.35 *		−0.13	0.04		−0.08	−0.20
	Preparation for surgery		0.15	0.15		0.01	−0.04		0.08	0.14
	Former hospitalizations		0.22	0.15		0.11	−0.04		0.14	0.24
3	Child’s problems			0.24 *			0.03			−0.04
	Parental stress reaction			0.02			0.22			−0.04
	PMTS	-	-	-			0.43 **			−0.44 *
	Medical phobia	-	-	-	-	-	-			−0.22 *
R^2^		0.05	0.34 **	0.38 **	0.04	0.13	0.33 *	0.06	0.10	0.46 *

**p* < 0.05, ** *p* < 0.01.

## Data Availability

The data presented in this study are available on request from the corresponding author. The data are not publicly available due to privacy constraints.

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
