# Peer review of "A Cross Sectional Study to Identify Traumatic Stress, Medical Phobia and Non-Adherence to Medical Care among Very Young Pediatric Patients"

_ijerph, 2023, doi:10.3390/ijerph20021122_

Round 1

Reviewer 1 Report

Thank you for the interesting article. While I see the importance of the topic, there are severe issues to it, which at the moment mostly seem to stem from a very messy approach to the writing. There is no goal or aim mentioned, there is a confusion of trauma and PTSD, mean values do not seem to make sense, sources are mentioned for the PTSD scale when the source refers to the PAT2.0, et cetera. However, there are also problems that might not be related to a messy writing of the text: the data seems to use children and parents at equal level in the analyses. For example: for the regressions, how does it work? Who is included in these regressions? The discussion is really not that reflective – in fact, there are very few sources in the article in general. It is also impossible for me at the moment to really assess the results, as certain values are recoded in a way that I cannot interpret them.

Due to the fact that 1, the introduction is incomplete, 2, the methods are unclear ,resulting in 3, unclear results, and thus 4, discussion which is very underdeveloped, I cannot recommend further review of this article. The authors should not lose heart over this, but take it as a chance to rework the article. The data is there, and I believe you can make a great article with it. But the article based on that data is simply not up to standard at the moment.

Below some more specific remarks:

The aim or hypotheses or objectives are absent from this study. It is not clear what this study actually wants to do. The introduction just stops at a certain point.

UCLA PTSD Reaction Index for DSM-5 Parent / Caregiver Version for Children Age 6 87 Years and Younger [22]: In order to assess post traumatic stress in both children and 88 adults, this measurement was first created in 1985. It possesses both high validity and 89 reliability to assess PTSD, among a variety of ethnic and religious groups [23]. Other 90 questionnaires of this type, the PTSD Checklist, PTSD Symptom Scale, and Harvard 91 Trauma Questionnaire, have similar results.

The DSM-5 was published in 2013. So, I believe the authors are mistaken when stating that this was created in 1985. PTSD inclusions were also completely different in 1985.

We found that severe trauma was related with high medical phobia (r = 0.23,

Is this different from “normal” medical phobia. Because now it seems as if the authors are assuming that the correlation indicates that it is mainly present for “high” medical phobia.

Secondly, the authors conflate “trauma” with “post-traumatic stress”. Trauma is your inclusion for PTSD, not a symptom in itself. So “severe trauma” refers to a lot of traumatic experiences, not to a scale.

I assume that the authors did not use criterion A1 for inclusion?

The authors mention two steps in the methods for inclusion. In the first step, parents complete questionnaires: which ones.

As for the second hypothesis in the study regarding medical phobia as a mediating fac- 138 tor of PMTS and adherence to medical treatment,

I did not realize actually that this study uses hypotheses. Where are these described?

I feel as if by accident the authors might have deleted an entire proportion of their introduction. Because they refer to “mediating factors”, but to my knowledge this is the first time this is even mentioned.

The UCLA measures PTSD, not trauma severity.

The confusion between trauma and PTSD leads to quite a confusing result section. So, if you state that trauma severity is correlated, do you mean the UCLA or the YCPC?

I don’t understand when the children completed any questionnaires?

I really don’t understand what happened in this study. If I’m correct, you have 2 groups: parents and children. But you seem to use them as one large dataset.

How can you, for example, correlated the UCLA and YCPC with each other? One is for parents, the other for children, right? So, these are different datasets. So, children will not have completed the UCLA, no? Or, is the UCLA measured in both? But what does the YCPC then measure? I assume the researcher did not measure PTSD twice in a row?

We recruited 152 parent and child pairs”- what are child pairs?

The mean values mentioned in table 2 do not make any sense to me. The M is very low – I don’t know what it means, because the methods do not mention how I should interpret the results. However, I looked it up myself, and if I’m not mistaken, the M mentioned here does not make any sense in relation to the scale. I assume there was some form of recoding.

It possesses both high validity and 89 reliability to assess PTSD, among a variety of ethnic and religious groups [23].

Source 23 refers to a scale called the PAT2.0 among children with cancer… I don’t see how that relates to the scale mentioned in the current article? PTSD is not even mentioned in that article.

You afterwards indeed refer to the PAT2.0 as scale, but do not mention the references there?

I would suggest also to clearly describe the research design.

Author Response

Author Response:  First off, thank you very much for your in-depth review of our article and recognizing the importance of this topic.  Thank you, as well, for taking the time review thoroughly and highlight the various problems within the paper.  You were absolutely correct. In the zeal to edit among various authors and make the initial changes requested by the journal editorial staff, we certainly lost the cohesion of the paper, and we agree with your assessment of its flaws. We have revised extensively, brought in much new research mentioned in the literature, and clarified both our goals and findings. We hope that this extensive revision will reflect the nature of our work in an acceptable fashion.

Below are individual responses to your specific comments.

Reviewer Comment: There is no goal or aim mentioned

Author Response:  Correct, and we added a paragraph at the end of the introduction.

Reviewer Comment: there is a confusion of trauma and PTSD,

Author Response:  Absolutely, therefore we clarified throughout the article the presence of post traumatic stress symptoms (PTSS) and our definition of PTSS in a medical setting for children, which has become known as Pediatric Medical Traumatic Stress  (PMTS)

Reviewer Comment: mean values do not seem to make sense

Author Response:  The author responsible for statistical analyses reviewed all of the data and corrected and clarified where necessary.

Reviewer Comment: sources are mentioned for the PTSD scale when the source refers to the PAT2.0.

Author Response: We clarified in the text that we used two distinct scales to measure PTSS/PTSD in children: the UCLA PTSD Reaction Index for DSM-5 Parent / Caregiver Version for Children Age 6 Years and Younger and the Young Child PTSD Checklist (YCPC).  Separately we applied two subscales of the PAT 2.0 to identify other variables that could affect medical phobia (MP), in child’s problems and parental stress reaction [variables named in the PAT 2.0 measurement scale]. 

Reviewer Comment: the data seems to use children and parents at equal level in the analyses. For example: for the regressions, how does it work? Who is included in these regressions?

Author Response: We clarified this in the procedures the results sections. The parent/caregiver completed all of the questionnaires, but reflected their child’s experiences/feelings/behaviors, where required.  This is the protocol when completing questionnaires on behalf of preschool aged children.

We also clarified this by adding this sentence in the procedures section:

As our study population was very young, the parents completed all the questionnaires based on their children’s responses and experiences.

Reviewer Comment: The discussion is really not that reflective – in fact, there are very few sources in the article in general

Author Response: This was accurate.  We went more in depth in our discussion section and included many more sources throughout the introduction and discussion to clarify/compare our findings.  All new sources are highlighted in the reference section.

Reviewer Comment: The aim or hypotheses or objectives are absent from this study. It is not clear what this study actually wants to do. The introduction just stops at a certain point.

Author Response: That was corrected with this paragraph:

Despite the growing body of research on PMTS and medical nonadherence among children, less is known about very young children and this association, especially after a medical intervention such as surgery.  Additionally, little is known about the role that medical phobia may play in the interaction.

The purpose of the study was to examine the consequences of traumatic medical events on the development of medical phobia in children. We hypothesized that PMTS symptoms would correlate with increased presence of a medical phobia and lower ad-herence to medical treatment.  We also hypothesized that this phobia would act as a me-diating factor of PMTS and adherence to medical treatment.

Reviewer Comment:  “UCLA PTSD Reaction Index for DSM-5 Parent / Caregiver Version for Children Age 6 87 Years and Younger [22]: In order to assess post traumatic stress in both children and 88 adults, this measurement was first created in 1985. It possesses both high validity and 89 reliability to assess PTSD, among a variety of ethnic and religious groups [23]. Other 90 questionnaires of this type, the PTSD Checklist, PTSD Symptom Scale, and Harvard 91 Trauma Questionnaire, have similar results.”

The DSM-5 was published in 2013. So, I believe the authors are mistaken when stating that this was created in 1985. PTSD inclusions were also completely different in 1985.

Author Response: Again, we apologize for the confusion.  We clarified this in the following sentences:

UCLA PTSD Reaction Index for DSM-5 Parent / Caregiver Version for Children Age 6 Years and Younger [31]: This measure possesses both high validity and reliability to as-sess PTSD, among a variety of ethnic and religious groups [23]. In this study, the internal reliability scale (Cronbach's α) was α = 0.95.

Reviewer Comment:  “We found that severe trauma was related with high medical phobia (r = 0.23,”Is this different from “normal” medical phobia. Because now it seems as if the authors are assuming that the correlation indicates that it is mainly present for “high” medical phobia.

Author Response: We corrected this to clarify that we are discussing lower or higher values on the scale that measures presence of medical fear/medical phobia.

As hypothesized, PTMS symptoms correlated with higher medical phobia scores (r = 0.23, p <0.01) and lower adherence to medical treatment (r = -0.46, p <0 .01). A moderate nega-tive correlation was found between the child’s medical phobia score and adherence to medical treatment (r = -0.22, p < 0.01). (Table 2)

Reviewer Comment:  Secondly, the authors conflate “trauma” with “post-traumatic stress”. Trauma is your inclusion for PTSD, not a symptom in itself. So “severe trauma” refers to a lot of traumatic experiences, not to a scale.

Author Response: Again, you are absolutely correct and we corrected this throughout the manuscript, that the we are measuring the psychological response to a possibly traumatic medical event. 

Reviewer Comment:  I assume that the authors did not use criterion A1 for inclusion?

Author Response: The study addresses Criterion A (as part of what is included within the UCLA PTSD Questionnaire). All the parents mentioned the child's surgery as an event perceived as dangerous and therefore meet this criterion.

Reviewer Comment:  The authors mention two steps in the methods for inclusion. In the first step, parents complete questionnaires: which ones.

Author Response: The We clarified this in the procedures section:

Stage 1) 152 parents completed the UCLA [31] and YCPC [3] questionnaires (identifying presence and level of PMTS). As our study population was very young, the parents com-pleted all the questionnaires based on their children’s responses and experiences.

Stage 2) Four months after discharge, 152 parents completed the UCLA [28] and YCPC [3] questionnaires (measuring PTMS), the CMFS [29] (level of fear/medical phobia), the PAT 2.0 (to measure overall child problems and caregiver stress) [32] and the Pittsburgh Questionnaire [30] to assess pediatric adherence to medical care.

Reviewer Comment:  “As for the second hypothesis in the study regarding medical phobia as a mediating fac- 138 tor of PMTS and adherence to medical treatment, “I did not realize actually that this study uses hypotheses. Where are these described?

Author Response: We corrected this by clarifying this in the goals paragraph at the end of the introduction:

The purpose of the study was to examine the consequences of traumatic medical events on the development of medical phobia in children. We hypothesized that PMTS symptoms would correlate with increased presence of a medical phobia and lower ad-herence to medical treatment.  We also hypothesized that this phobia would act as a me-diating factor of PMTS and adherence to medical treatment.

Reviewer Comment:  I feel as if by accident the authors might have deleted an entire proportion of their introduction. Because they refer to “mediating factors”, but to my knowledge this is the first time this is even mentioned.

Author Response: This indeed was the case.  We inserted the correct paragraph and elaborated on this issue to clarify both what we meant by mediating factors and the importance of this idea.

Medical phobia may be another clinical manifestation of the traumatic medical event, appearing with great frequency with PMTS [23]. In fact, medical phobia may be a factor that explains the connection between a traumatic medical event and medical adherence [24]. Phobia is defined as intense and irrational anxiety or fear of a specific object or situation that evokes in the individual a powerful and disproportionate fear [25]. Medical phobia is defined as fear of exposure to a medical procedure (e.g., injection [26] medical examination, or dental treatment [27] or fear of injury [25] or even fever in a child [28].

Reviewer Comment:  The UCLA measures PTSD, not trauma severity.

Author Response: Absolutely and this was clarified in the text as mentioned previously.

Reviewer Comment:  The confusion between trauma and PTSD leads to quite a confusing result section. So, if you state that trauma severity is correlated, do you mean the UCLA or the YCPC?

Author Response: As shown in table number 2, the analysis of the correlation was done on both questionnaires in order to achieve convergent validity. We  found that with both measurements there was a correlation between the severity of PTMS symptoms and the level of phobia.

To clarify this concept, we added this sentence in the procedures section:

Stage 1) 152 parents completed the UCLA [31] and YCPC [3] questionnaires (identifying presence and level of PMTS). Both questionnaires were used for analysis in order to achieve convergent validity. As our study population was very young, the parents com-pleted all the questionnaires based on their children’s responses and experiences.

Reviewer Comment:  I don’t understand when the children completed any questionnaires?

Author Response: As clarified above, the parent completed the questionnaires on behalf of the child.  To clarify time line we added these paragraphs to the procedures section?

After receiving ethics approval from the sponsoring organization, during consecutive hospitalizations of 204 children from April to July 2018, 184 parents agreed to participate in the study.  An explanation of the study was provided and parents signed a consent form and completed the questionnaires with the help of researchers during a face-to-face interview. Six parents refused to participate because of the emotional costs; 28 parents cited lack of time for their refusal.

Stage 1) 152 parents completed the UCLA [31] and YCPC [3] questionnaires (identifying presence and level of PMTS). Both questionnaires were used for analysis in order to achieve convergent validity. As our study population was very young, the parents com-pleted all the questionnaires based on their children’s responses and experiences.

Stage 2) Four months after discharge, 152 parents completed the UCLA [28] and YCPC [3] questionnaires (measuring PTMS), the CMFS [29] (level of fear/medical phobia), the PAT 2.0 (to measure overall child problems and caregiver stress) [32] and the Pittsburgh Ques-tionnaire [30] to assess pediatric adherence to medical care.

Reviewer Comment:  I really don’t understand what happened in this study. If I’m correct, you have 2 groups: parents and children. But you seem to use them as one large dataset.

How can you, for example, correlated the UCLA and YCPC with each other? One is for parents, the other for children, right? So, these are different datasets. So, children will not have completed the UCLA, no? Or, is the UCLA measured in both? But what does the YCPC then measure? I assume the researcher did not measure PTSD twice in a row?

Author Response: We apologize that our paper was not clear on this issue.  We hope that we have sufficiently clarified now in previous responses.  We utilized two questionnaires that both measure PTSD symptoms in very young children in order to create a convergent validity (testing our hypotheses that there was indeed a correlation between PTMS and medical phobia, by analyzing correlation of each measurement [UCLA and YCPC] separately.  Additionally, the parents completed the questions on behalf of their child.

Reviewer Comment:  “We recruited 152 parent and child pairs”- what are child pairs?

Author Response: We clarified this in the following sentences:

For this analysis, we included 152 parent and child dyads (one parent completing the questionnaire on behalf of themselves and their child) who completed the questionnaires at both stages. 

Reviewer Comment:  The mean values mentioned in table 2 do not make any sense to me. The M is very low – I don’t know what it means, because the methods do not mention how I should interpret the results. However, I looked it up myself, and if I’m not mistaken, the M mentioned here does not make any sense in relation to the scale. I assume there was some form of recoding.

Author Response: This was corrected and table 2 now and the M values should now make sense.

M

SD

1

2

3

1. UCLA PTSS presence

7.82

10.87

2. YCPC PTSS presence

7.73

9.69

0.90**

3. Child’s medical phobia

1.60

0.60

0.23**

0.24**

4. Child’s adherence to medical treatment

3.76

1.89

-0.46**

-0.49**

-0.22**

Reviewer Comment:  “It possesses both high validity and 89 reliability to assess PTSD, among a variety of ethnic and religious groups [23].” Source 23 refers to a scale called the PAT2.0 among children with cancer… I don’t see how that relates to the scale mentioned in the current article? PTSD is not even mentioned in that article. You afterwards indeed refer to the PAT2.0 as scale, but do not mention the references there?

Author Response: We clarified this confusion and added these sentences to the manuscript.

Psychosocial Assessment Tool II (PAT 2.0) [32] designed to assess psychosocial problems in children with life-threatening diseases. The instrument includes 84 items and is com-prised of 7 sub-scales: Family Structure and Resources, Family Social Support, Family Problems, Parent Stress Reaction, Family Beliefs, Child Problems, and Sibling Problems. Each item in the sub-scales is dichotomously scored. Higher scores are associated with higher levels of psychosocial risk for the child. For this analysis, we utilized the subscales of Child Problems and Parent Stress Reaction.

As well as in the results section:

Variables were entered in three steps. In step 1, we entered the child’s gender and age; in step 2, we entered clinical and hospitalization variables. In step 3, we entered the child’s problems score, parental stress reaction score, the UCLA scores that measures PMTS and medical phobia scores (in separate regressions).

And

Finally, as found in step 3 of regression, child’s problems score was positively related with PMTS symptom severity (β = 0.24, p < 0.01) but parental stress reaction was not.

We also clarified this in Table 3 using the correct subset category names of the PAT 2.0 [child’s problems and parental stress reaction scales]

Reviewer Comment:  I would suggest also to clearly describe the research design.

Author Response: Below is the revised procedures section which we hope clarifies our research design:

After receiving ethics approval from the sponsoring organization, during consecutive hospitalizations of 204 children from April to July 2018, 184 parents agreed to participate in the study.  An explanation of the study was provided and parents signed a consent form and completed the questionnaires with the help of researchers during a face-to-face interview. Six parents refused to participate because of the emotional costs; 28 parents cited lack of time for their refusal.

Stage 1) 152 parents completed the UCLA [31] and YCPC [3] questionnaires (identifying presence and level of PMTS). Both questionnaires were used for analysis in order to achieve convergent validity. As our study population was very young, the parents com-pleted all the questionnaires based on their children’s responses and experiences.

Stage 2) Four months after discharge, 152 parents completed the UCLA [28] and YCPC [3] questionnaires (measuring PTMS), the CMFS [29] (level of fear/medical phobia), the PAT 2.0 (to measure overall child problems and caregiver stress) [32] and the Pittsburgh Ques-tionnaire [30] to assess pediatric adherence to medical care.

Reviewer 2 Report

The current study examined the relationships between parent report of traumatic stress reaction, medical phobia symptoms, and non-adherence to medical care among very young pediatric patients following hospitalization in a surgical ward. The topics covered are very important and timely. Strengths of the study include a moderate sample size, use of several well-validated questionnaire measures, and discussion of clinical implications. There are also several weaknesses of the study that are described below and should be addressed prior to publication.

Introduction: Discussion of the overlaps and distinctions between PMTS and Medical Phobia is necessary in the Introduction. Relatedly, the authors can further clarify the precise goal of this study as at present it is not clear from the Introduction section. Further, the Introduction section can be improved by adding discussion of the large body of literature on the association between PTSS symptoms and medical non-adherence in both adults and children. In particular, inclusion of more recent and larger studies as well as reviews is appropriate.

Materials and Methods: Since the Pittsburgh Rehabilitation Participation Scale was only completed by parents in this study, it is confusing to state that it is completed by caregivers, parents, and professionals in the initial description. Please include the number of items for each questionnaire. Step 1 of the Procedure does not specify which questionnaires were completed whereas Step 2 does. It is unclear when the PAT 2.0 was administered. As I cannot tell exactly which measures were administered when based on the current manuscript draft, I am unsure whether the mediation analysis is appropriately interpreted. Generally, causality cannot be assumed when the predictor, mediator, and/or outcome are measured at the same time-point. Also, it is unclear if the authors controlled for multiple comparisons when running analyses.  

Results: The results state that "severe trauma was related with high medical phobia and low adherence to medical treatment." Are these descriptors (severe, high, low) based on clinical cut-offs? Later when medical phobia is discussed, the phrase “the presence of medical phobia” is used which suggests the measure was dichotomized (i.e., present, not present), but this is not described. In another place, the wording “pediatric medical traumatic stress syndrome” is used but it is not stated what qualifies as syndromic. It is also unclear what “children’s mental difficulties” and “parent’s mental difficulties” refer to. I believe they were derived from the PAT 2.0, but the labels differ from the subscales listed in the methods section and it was not indicated which subscales were investigated and why those were selected.

In regard to the relationship between trauma reaction severity and the medical phobia, these two measures are significantly correlated which may negatively affect their interpretability in the regression model predicting adherence. This seems to be the case. Although both trauma reaction severity and medical phobia are positively correlated with each other and both negatively correlated with adherence, the signs of the betas in the regression do not match.

The authors state that the results indicate that medical phobia mediated the relationship between trauma reaction severity and adherence, indicating that trauma reaction severity leads to stronger medical phobia, which, in turn, leads to lower pediatric adherence to medical treatment. However, this interpretation cannot be drawn from the current study. For hierarchical regression, medical phobia and trauma reaction severity would need to be included in separate steps of the hierarchical regression predicting adherence to assess for statistical mediation. For SEM, in a full mediation process, the effect is 100% mediated by the mediator, that is, in the presence of the mediator, the pathway connecting the predictor to the outcome is completely broken so that the predictor has no direct effect on the outcome. However, the authors have stated that their model found a full mediation effect as well as direct and indirect effects. Further, to draw causal conclusions such as trauma reaction severity leads to medical phobia which leads to lower treatment adherence, these variables would need to be measured at three different time points. The Measures section indicates that the CMFS was measured at multiple timepoints, but it is unclear from the Data Analysis and Results section which timepoint was used. It is also unclear if the UCLA and YCPC were administered at time 1 and if these data were used in analyses. Also, the R-squared for the regression predicting trauma severity (UCLA) is larger than that of the regression predicting medical phobia. Could this suggest that medical phobia is not distinct from trauma severity in this sample?

Discussion: As I described above, the paper overstates the associations found and inappropriately concludes causality. The way these relationships are described should be adjusted accordingly. Also, the authors state that trauma-related avoidance results in medical nonadherence, but based on the data presented in the paper, this is a postulation, and the language should be tempered to indicate that this is not a firm conclusion. Avoidance as a trauma reaction in particular is not explored and the causality of this claim is not supported by the data.

The authors also make an argument for including PMTS as a new diagnosis and provide strong clinical application recommendations. They can better tie in the results of this study into these arguments.

Additionally, given that the focus of this study is on very young children whose caregivers are primarily responsible to administering medications and bringing children to appointments, more attention should be given to their role in managing adherence and how this may be impacted by their children’s or their own mental health. The study’s findings could be compared to those with older samples.

Limitations: The authors can discuss the fact that all measures are parent-report and therefore subject to reporter bias including parental stress. Additionally, since traumatic stress reaction and medical phobia were both measured at the same time as treatment adherence, inferences about causality cannot be made.

Editorial errors: The paper should be reviewed for English language and style as some sentences are unclear. Also when discussing trauma reactions, it is more appropriate to say “trauma reaction severity” than “trauma severity” (i.e., lines 150, 152). Further, there are erroneously capitalized words, misplaced commas, erroneously bolded words, missing articles, and other minor errors (i.e., centered text in 1st column of Table 1) throughout the manuscript.

Author Response

The current study examined the relationships between parent report of traumatic stress reaction, medical phobia symptoms, and non-adherence to medical care among very young pediatric patients following hospitalization in a surgical ward. The topics covered are very important and timely. Strengths of the study include a moderate sample size, use of several well-validated questionnaire measures, and discussion of clinical implications. There are also several weaknesses of the study that are described below and should be addressed prior to publication.

Reviewer Comment:   Introduction: Discussion of the overlaps and distinctions between PMTS and Medical Phobia is necessary in the Introduction. Relatedly, the authors can further clarify the precise goal of this study as at present it is not clear from the Introduction section. Further, the Introduction section can be improved by adding discussion of the large body of literature on the association between PTSS symptoms and medical non-adherence in both adults and children. In particular, inclusion of more recent and larger studies as well as reviews is appropriate.

Author Response: Thank you for this clarification.  We reviewed and included more research on the association between PTSS symptoms and medical non-adherence in both adults and children.  The added references are highlighted in the reference section and we also include them here.

  1. 17. Pai, A. L.; McGrady, M. Systematic review and meta-analysis of psychological in-terventions to promote treatment adherence in children, adolescents, and young adults with chronic illness. J Pediatr Psychol 2014 39, 918–931.
  2. 18. Armstrong, M.L.; Duncan, C.L.; Stokes, J.O.; Pereira, D. Association of caregiver health beliefs and parenting stress with medication adherence in preschoolers with asthma. J Asthma 2014 51(4), 366-372.
  3. 19. Plevinsky, J.M.; Gutierrez-Colina, A.M.; Carmody, J.K.; Hommel, K.A.; Crosby, L.E.; McGrady, M.E.; Pai, A.L.; Ramsey, R.R.; Modi, A.C. Patient-reported outcomes for pediatric adherence and self-management: A systematic review. J Pediatr Psychol 2020 45(3), 340-357.
  4. 20. Killian, M.O.; Schuman, D.L.; Mayersohn, G.S.; Triplett, K.N. Psychosocial pre-dictors of medication non‐adherence in pediatric organ transplantation: A sys-tematic review. Pediatr Transplant 2018 22(4), e13188.
  5. Duncan‐Park, S.; Danziger‐Isakov, L.; Armstrong, B.; Williams, N.; Odim, J.; Shemesh, E.; Sweet, S.; Annunziato, R. Posttraumatic stress and medication ad-herence in pediatric transplant recipients. Am J Transplant 2022 22(3), 937-946.

Reviewer Comment:   Materials and Methods: Since the Pittsburgh Rehabilitation Participation Scale was only completed by parents in this study, it is confusing to state that it is completed by caregivers, parents, and professionals in the initial description. Please include the number of items for each questionnaire. Step 1 of the Procedure does not specify which questionnaires were completed whereas Step 2 does. It is unclear when the PAT 2.0 was administered. As I cannot tell exactly which measures were administered when based on the current manuscript draft, I am unsure whether the mediation analysis is appropriately interpreted. Generally, causality cannot be assumed when the predictor, mediator, and/or outcome are measured at the same time-point. Also, it is unclear if the authors controlled for multiple comparisons when running analyses.

Author Response: We clarified this in the procedures section:

After receiving ethics approval from the sponsoring organization, during consecutive hospitalizations of 204 children from April to July 2018, 184 parents agreed to participate in the study.  An explanation of the study was provided and parents signed a consent form and completed the questionnaires with the help of researchers during a face-to-face interview. Six parents refused to participate because of the emotional costs; 28 parents cited lack of time for their refusal.

Stage 1) 152 parents completed the UCLA [31] and YCPC [3] questionnaires (identifying presence and level of PMTS). Both questionnaires were used for analysis in order to achieve convergent validity. As our study population was very young, the parents com-pleted all the questionnaires based on their children’s responses and experiences.

Stage 2) Four months after discharge, 152 parents completed the UCLA [28] and YCPC [3] questionnaires (measuring PTMS), the CMFS [29] (level of fear/medical phobia), the PAT 2.0 (to measure overall child problems and caregiver stress) [32] and the Pittsburgh Ques-tionnaire [30] to assess pediatric adherence to medical care.

Additionally, we corrected the wording regarding predicting, ensuring that we only referred to correlations, since causality couldn’t be determined by our study.  We apologize for the confusion.

Reviewer Comment:   Results: The results state that "severe trauma was related with high medical phobia and low adherence to medical treatment." Are these descriptors (severe, high, low) based on clinical cut-offs? Later when medical phobia is discussed, the phrase “the presence of medical phobia” is used which suggests the measure was dichotomized (i.e., present, not present), but this is not described. In another place, the wording “pediatric medical traumatic stress syndrome” is used but it is not stated what qualifies as syndromic. It is also unclear what “children’s mental difficulties” and “parent’s mental difficulties” refer to. I believe they were derived from the PAT 2.0, but the labels differ from the subscales listed in the methods section and it was not indicated which subscales were investigated and why those were selected.

Author Response: We clarified all of our references to severe trauma to the correct term of PTSS or PMTS. We also clarified that issue of high and low medical phobia. We also clarified PMTS throughout the manuscript, as well as the use of “children and parent mental difficulties”, which refers to the child’s problems and parental stress reaction subscales of the PAT 2.0 . 

Psychosocial Assessment Tool II (PAT 2.0) [32] designed to assess psychosocial problems in children with life-threatening diseases. The instrument includes 84 items and is com-prised of 7 sub-scales: Family Structure and Resources, Family Social Support, Family Problems, Parent Stress Reaction, Family Beliefs, Child Problems, and Sibling Problems. Each item in the sub-scales is dichotomously scored. Higher scores are associated with higher levels of psychosocial risk for the child. For this analysis, we utilized the subscales of Child Problems and Parent Stress Reaction.

In regard to the relationship between trauma reaction severity and the medical phobia, these two measures are significantly correlated which may negatively affect their interpretability in the regression model predicting adherence. This seems to be the case. Although both trauma reaction severity and medical phobia are positively correlated with each other and both negatively correlated with adherence, the signs of the betas in the regression do not match.

Reviewer Comment:   The authors state that the results indicate that medical phobia mediated the relationship between trauma reaction severity and adherence, indicating that trauma reaction severity leads to stronger medical phobia, which, in turn, leads to lower pediatric adherence to medical treatment. However, this interpretation cannot be drawn from the current study. For hierarchical regression, medical phobia and trauma reaction severity would need to be included in separate steps of the hierarchical regression predicting adherence to assess for statistical mediation. For SEM, in a full mediation process, the effect is 100% mediated by the mediator, that is, in the presence of the mediator, the pathway connecting the predictor to the outcome is completely broken so that the predictor has no direct effect on the outcome. However, the authors have stated that their model found a full mediation effect as well as direct and indirect effects. Further, to draw causal conclusions such as trauma reaction severity leads to medical phobia which leads to lower treatment adherence, these variables would need to be measured at three different time points. The Measures section indicates that the CMFS was measured at multiple timepoints, but it is unclear from the Data Analysis and Results section which timepoint was used. It is also unclear if the UCLA and YCPC were administered at time 1 and if these data were used in analyses. Also, the R-squared for the regression predicting trauma severity (UCLA) is larger than that of the regression predicting medical phobia. Could this suggest that medical phobia is not distinct from trauma severity in this sample?

Author Response: You were absolutely correct and this has been revised extensively in this manuscript.

Reviewer Comment:   Discussion: As I described above, the paper overstates the associations found and inappropriately concludes causality. The way these relationships are described should be adjusted accordingly. Also, the authors state that trauma-related avoidance results in medical nonadherence, but based on the data presented in the paper, this is a postulation, and the language should be tempered to indicate that this is not a firm conclusion. Avoidance as a trauma reaction in particular is not explored and the causality of this claim is not supported by the data.

Author Response: we corrected the wording regarding predicting, ensuring that we only referred to correlations, since causality couldn’t be determined by our study.  We apologize for the confusion.

Reviewer Comment:   The authors also make an argument for including PMTS as a new diagnosis and provide strong clinical application recommendations. They can better tie in the results of this study into these arguments.

Author Response: We strengthened this argument in our discussion section with support from the literature and a clearer argument.  A sample of this would be:

Traumatic medical experiences and the consequent development of medical phobia may lead to the development of avoidance patterns and the child’s refusal to cooperate with future treatments and examinations. This has implications, both for the individual patient and for the system in general. Specifically, the child’s refusal to cooperate in future treatments and medical examinations could impair the child's rehabilitative process, with negative prognostic consequences. This impact has been documented among pediatric oncology patients who have reduced adherence to important chemotherapy medications [36], as well as young children with asthma [18] and other chronic illness [19]. Systemically, lack of adherence affects the entire health system, adding high financial costs to the system [39]. Therefore, there is a need to minimize possible pediatric traumatic experience during hospitalization and swiftly identify possible sequelae of PMTS and medical phobia.

Reviewer Comment:    Additionally, given that the focus of this study is on very young children whose caregivers are primarily responsible to administering medications and bringing children to appointments, more attention should be given to their role in managing adherence and how this may be impacted by their children’s or their own mental health. The study’s findings could be compared to those with older samples.

 Author Response: In order to strengthen our argument, we include many new studies on medical adherence on preschool aged children

  1. 18. Armstrong, M.L.; Duncan, C.L.; Stokes, J.O.; Pereira, D. Association of caregiver health beliefs and parenting stress with medication adherence in preschoolers with asthma. J Asthma 2014 51(4), 366-372.
  2. 19. Plevinsky, J.M.; Gutierrez-Colina, A.M.; Carmody, J.K.; Hommel, K.A.; Crosby, L.E.; McGrady, M.E.; Pai, A.L.; Ramsey, R.R.; Modi, A.C. Patient-reported outcomes for pediatric adherence and self-management: A systematic review. J Pediatr Psychol 2020 45(3), 340-357.
  3. Killian, M.O.; Schuman, D.L.; Mayersohn, G.S.; Triplett, K.N. Psychosocial pre-dictors of medication non‐adherence in pediatric organ transplantation: A sys-tematic review. Pediatr Transplant 2018 22(4), e13188.

Reviewer Comment:    Limitations: The authors can discuss the fact that all measures are parent-report and therefore subject to reporter bias including parental stress. Additionally, since traumatic stress reaction and medical phobia were both measured at the same time as treatment adherence, inferences about causality cannot be made.

Author Response: We revised the limitation section to include this bias:

Our study took place in only one hospital ward. Further research should focus on medical centers and additional populations in order to expand the external validity of the findings. However, the sample in the study included a wide range of children undergoing different types of surgical hospitalizations. Moreover, the sample included a wide range of population characteristics (socioeconomically), which does increase the applicability to other populations. Additionally, we assessed, adherence to medical care as the care parents provided the child at home. This type of care is fundamentally different than assessment of adherence to hospital care. When a child receives home treatment, there is moderation of medical phobia that may reduce the avoidance. As we examined the behaviors and experiences of preschool aged children, we also have the possible bias of parental reporting on their child’s thoughts and feelings, which may not accurately reflect the child’s actual perceptions.

We also removed mention of causality throughout the manuscript, and clarified that our results only analyzed correlation.

Reviewer Comment:    Editorial errors: The paper should be reviewed for English language and style as some sentences are unclear. Also when discussing trauma reactions, it is more appropriate to say “trauma reaction severity” than “trauma severity” (i.e., lines 150, 152). Further, there are erroneously capitalized words, misplaced commas, erroneously bolded words, missing articles, and other minor errors (i.e., centered text in 1st column of Table 1) throughout the manuscript.

Author Response: The manuscript was re-edited and reviewed by a professional editor who is a native English speaker.

Round 2

Reviewer 1 Report

I thank the authors for the extensive reworking of the article.

Author Response

As suggested by the reviewer, editing was done at the linguistic level

Reviewer 2 Report

I appreciate the authors’ commitment to improving this manuscript. They have clearly made substantial efforts to respond to the suggestions. The manuscript is much improved overall. However, there are several points that have not been adequately addressed and, in my opinion, the manuscript should be further revised prior to publication.

1.     The language in the Abstract (i.e., “indicating that PMTS severity leads to stronger medical phobia, which, in turn, leads to lower pediatric adherence to medical treatment.”) is still causal and should be modified.

2.     The introduction has been much improved. One note though: ethnic minority status itself is not a reason for medical nonadherence, though it may be a proxy for something else like lower SES or access to resources, lower health literacy and/or language/technological barriers to accessing medical guidance, or lower trust in the medical establishment. Wording should be adjusted.

3.     Within the Methods, I still advise the authors to include the number of items for each questionnaire.

4.     Again, the Pittsburgh scale description still says it is completed by the child’s caregivers, parents, and professionals, yet in this study it was only parents.

5.     How was SES measured? The methods describe collecting data on parent education level, but Table 1 simply states SES.

6.     Was the PMTS score collected at stage 1 or 2 used in the regressions and correlations run with medical phobia scores and adherence?

7.     Typically, statistical analytic plans are included in the methods section, but in the draft they are in the results section. Consider revising.

8.     In Table 2, the use of PTSS “presence” is confusing because it is a dimensional scale. I think PTSS severity, level, or score would be more appropriate.

9.     In Table 3, the “trauma severity (UCLA)” heading should be adjusted to PMTS severity (UCLA).

10.  In my previous review I suggested how the stepwise regression could be re-run to match the described goal of this manuscript; however, the analyses have not been adjusted. I am confused what the purpose of these hierarchical regressions is as they currently are described. Were these analyses conducted to determine appropriate co-variates for the SEM analysis? If so, this is unclear and not stated.

11.  The SEM results are the same but described as showing a partial mediation (rather than full, as in the previous draft). Based on the manuscript and the author’s response I am unsure whether the initial draft just included a typo and did not necessitate re-analysis.

12.  The Discussion section can be further strengthened to emphasize precisely what this study has added to the literature. At present, it is not entirely clear.

13.  Please address my previous comments:

a.     It is unclear if the authors controlled for multiple comparisons when running analyses.

b.     In regard to the relationship between trauma reaction severity and the medical phobia, these two measures are significantly correlated which may negatively affect their interpretability in the regression model predicting adherence. This seems to be the case. Although both trauma reaction severity and medical phobia are positively correlated with each other and both negatively correlated with adherence, the signs of the betas in the regression do not match.

c.     The R-squared for the regression predicting trauma severity (UCLA) is larger than that of the regression predicting medical phobia. Could this suggest that medical phobia is not distinct from trauma severity in this sample?

d.     The authors also make an argument for including PMTS as a new diagnosis and provide strong clinical application recommendations. They can better tie in the results of this study into these arguments.

                                               i.     I appreciate that the authors added in support from the literature, but encourage them to tie in their results better.

e.     Additionally, given that the focus of this study is on very young children whose caregivers are primarily responsible to administering medications and bringing children to appointments, more attention should be given to their role in managing adherence and how this may be impacted by their children’s or their own mental health.

                                               i.     The added citations are not about the role of parents.

f.      Limitations: The authors can discuss the fact that all measures are parent-report and therefore subject to reporter bias including parental stress.

                                               i.     In the response, the authors indicated that they added the following that is not included in the attached manuscript: “As we examined the behaviors and experiences of preschool aged children, we also have the possible bias of parental reporting on their child’s thoughts and feelings, which may not accurately reflect the child’s actual perceptions.” This omission should be included. It may also be prudent to recognize the limitations inherent to a single informant providing all outcome variables rather than multiple informants/collection methods (i.e., reporting bias, response style, impact of the parents’ own mental health on their ratings).

14.  Generally, the article read much better now, though there remain a few typos:

a.     PAT 2.0 description – first sentence is incomplete.

b.     Single vs. double spacing between sentences – inconsistent

c.     Need a comma between “gender of the child” and “gender of the participating parent” – line 80

d.     Need to capitalize “parents” in line 119

e.     Incomplete sentence – line 132-133

f.      Multiple types on the 2nd sentence of the discussion (lines 190-193)

Author Response

Response to comments and suggestions of article

A cross sectional study to identify traumatic stress, medical phobia and non-adherence to medical care among very young pediatric patients

Reviewer Comment: The language in the Abstract (i.e., “indicating that PMTS severity leads to stronger medical phobia, which, in turn, leads to lower pediatric adherence to medical treatment.”) is still causal and should be modified.

Author response: This was changed in the current manuscript to read “In addition, MP mediated the relationship between PMTS severity and adherence, indicating that PMTS severity is associated with stronger medical phobia, and lower pediatric adherence to medical treatment.”

Reviewer Comment: ethnic minority status itself is not a reason for medical nonadherence, though it may be a proxy for something else like lower SES or access to resources, lower health literacy and/or language/technological barriers to accessing medical guidance, or lower trust in the medical establishment. Wording should be adjusted.

Author response: Despite several reviews which have identified minority ethnic status as a risk factor for pediatric medical adherence without specifying it as a proxy, we do understand the confusion and therefore have removed its mention from the introduction.

Reviewer Comment: Within the Methods, I still advise the authors to include the number of items for each questionnaire.

Author response:  Pittsburgh Rehabilitation Participation Scale consists of 2 items.  The UCLA PTSD Reaction Index for DSM-5 Parent / Caregiver Version for Children Age 6 Years and Younger consists of 22 items.  These details were added to the measures descriptions in the Methods section.  

Reviewer Comment: Again, the Pittsburgh scale description still says it is completed by the child’s caregivers, parents, and professionals, yet in this study it was only parents.

Author response:  Changed to This questionnaire was completed by the child's parents

Reviewer Comment: How was SES measured? The methods describe collecting data on parent education level, but Table 1 simply states SES.

Author response:  Socio-economic status is tested by weighting the subject's answers to the following questions: years of education of the parents, number of rooms in the house, and presence of any financial difficulties  (self report).  This explanation was added to the measures section.

Reviewer Comment: Was the PMTS score collected at stage 1 or 2 used in the regressions and correlations run with medical phobia scores and adherence?

Author response:  We used the score from Stage 2 in the regressions. To clarify this, we added the following to the Data Analysis section, “and used variable scores generated from Stage 2 of the study only.”.

Reviewer Comment: Typically, statistical analytic plans are included in the methods section, but in the draft they are in the results section. Consider revising.

Author response:  Data analysis is at the end of the methods section

Reviewer Comment: In Table 2, the use of PTSS “presence” is confusing because it is a dimensional scale. I think PTSS severity, level, or score would be more appropriate.

Author response: PTSS “presence” changed to severity

Reviewer Comment: In Table 3, the “trauma severity (UCLA)” heading should be adjusted to PMTS severity (UCLA).

Author response: “trauma severity (UCLA)” changed to PTSS severity (UCLA)

Reviewer Comment: In my previous review I suggested how the stepwise regression could be re-run to match the described goal of this manuscript; however, the analyses have not been adjusted. I am confused what the purpose of these hierarchical regressions is as they currently are described. Were these analyses conducted to determine appropriate co-variates for the SEM analysis? If so, this is unclear and not stated.

Author response: The presented hierarchical regressions aimed to provide information regarding the unique contribution of three sets of variables to the explanation of outcomes: (1) demographics. (2) surgery's related variables and (3) psychological variables. To clarify we added this sentence to the results section.

Reviewer Comment: The SEM results are the same but described as showing a partial mediation (rather than full, as in the previous draft). Based on the manuscript and the author’s response I am unsure whether the initial draft just included a typo and did not necessitate re-analysis.

Author response: Yes, the initial draft had a typo that was corrected and therefore did not necessitate re-analysis.

Reviewer Comment: The Discussion section can be further strengthened to emphasize precisely what this study has added to the literature. At present, it is not entirely clear.

Author response:  This paragraph was added to the conclusion to clarify how this study contributes to the literature.  “The findings of the present study indicate that medical phobia is positively and significantly related to PMTS and that it is a mediating factor between medical hospitalization and compliance with treatment. This study is an important addition to the research literature due to the fact that it specifically examines the consequences of traumatic distress on the patient's level of compliance to attend follow-up treatments and cooperate with the treatment plan. Since it is known that a lack of cooperation is a significant barrier to recovery, there is significant value in the findings of the current study that shed light on the understanding of the dynamics that lead to this pattern.”

Reviewer Comment: It is unclear if the authors controlled for multiple comparisons when running analyses.

Author response: We did, and to clarify that, we added this sentence in the data analysis section. “We controlled for multiple comparisons.”

Reviewer Comment: In regard to the relationship between trauma reaction severity and the medical phobia, these two measures are significantly correlated which may negatively affect their interpretability in the regression model predicting adherence. This seems to be the case. Although both trauma reaction severity and medical phobia are positively correlated with each other and both negatively correlated with adherence, the signs of the betas in the regression do not match.

Specifically, PMTS was associated with a stronger presence of medical phobia (β = 0.14, p < 0.05), which, in turn, was negatively correlated with the child’s reduced adherence to medical treatment (β = -0.22, p < 0.01). We apologize for the confusion, but in both tables and the results section, the betas are both negative. The beta is negative because the stronger the trauma reaction the less adherence.  To clarify the relation, we added negative correlation in the sentence, “Specifically, PMTS was associated with a stronger presence of medical phobia (β = 0.14, p < 0.05), which, in turn, was negatively correlated with the child’s reduced adherence to medical treatment (β = -0.22, p < 0.01).”

Reviewer Comment: The R-squared for the regression predicting trauma severity (UCLA) is larger than that of the regression predicting medical phobia. Could this suggest that medical phobia is not distinct from trauma severity in this sample?

Author response: This is indeed an important question, but in our opinion, this possibility is unlikely because these are two distinct variables that test different consequences of a potentially traumatic medical event.

Reviewer Comment: The authors also make an argument for including PMTS as a new diagnosis and provide strong clinical application recommendations. They can better tie in the results of this study into these arguments.

Author response: We inserted this into our discussion to better tie in our results.

“The diagnosis of PMTS refers to the symptoms of post-traumatic stress disorder that developed following an illness or medical hospitalization. The findings of the current study show that the development of medical phobia and the subsequent impairment in compliance with reaching treatment are part of the consequences of such an event. Because of this, the researchers' proposal is to include the components of medical phobia and avoidance of treatment as part of the definition of medical phobia. These components are unique to PMTS and different from what happens in other cases of PTSD, so it is important to consider them when diagnosing PMTS.”

Reviewer Comment: Additionally, given that the focus of this study is on very young children whose caregivers are primarily responsible to administering medications and bringing children to appointments, more attention should be given to their role in managing adherence and how this may be impacted by their children’s or their own mental health.   The added citations are not about the role of parents.

Author response: This is a correct point. Indeed, for preschool children, the degree of their medical adherence is dependent on the parents because usually a 3-year-old child is not asked if he wants to go to the doctor, but simply taken. However, when a child throws a tantrum before any medical examination, for some parents, it will have an effect and they may avoid taking the child to follow-up treatments/follow the rehabilitation plan because they do not have the strength to stand up to the child. 

In the article we added a sentence on the importance of instructing parents on how to conduct themselves in such cases is extremely important.    “In terms of adherence, our study was only able to focus on medication taking. Be-cause our population centered on preschool aged children, we recognize that the degree of the child’s medical adherence is strongly related to the parents. However, when a child throws a tantrum before a medical examination, or refuses to take medication, some par-ents may not know how to cope with these refusals, and therefore the child will not adhere to the treatment regime [36-37]. Specifically for these parents, behavioral modification in-terventions may be helpful, but further investigation is needed to determine specific barri-ers to adherence among this subset of preschool children [41]”

However, the present article is not concerned with providing a solution to the problem of adherence, but mainly to point out the problem and the dynamics that created it (surgery - medical phobia - reduced response to treatment) and the importance of understanding this dynamic to understanding the existence and repercussions of PMTS.

Reviewer Comment: Limitations: The authors can discuss the fact that all measures are parent-report and therefore subject to reporter bias including parental stress.  In the response, the authors indicated that they added the following that is not included in the attached manuscript: “As we examined the behaviors and experiences of preschool aged children, we also have the possible bias of parental reporting on their child’s thoughts and feelings, which may not accurately reflect the child’s actual perceptions.” This omission should be included.

Author response: This sentence appears at the end of the limitations section but was not highlighted so perhaps it was missed.

Reviewer Comment: It may also be prudent to recognize the limitations inherent to a single informant providing all outcome variables rather than multiple informants/collection methods (i.e., reporting bias, response style, impact of the parents’ own mental health on their ratings).

Author response: This sentence was added to the limitations section, “Also, there is an inherent bias when a single reports  all outcome variables rather than multiple informants/collection methods  which would also affect our results.”

Reviewer Comment: Generally, the article read much better now, though there remain a few typos: PAT 2.0 description – first sentence is incomplete.

Author response: Sentence was corrected to read, “This measure is designed to assess psychosocial problems in children with life-threatening diseases.”

Reviewer Comment: Single vs. double spacing between sentences – inconsistent

Author response:  We believe this is the formatting of the journal as we have not changed spacing.”

Reviewer Comment: Need a comma between “gender of the child” and “gender of the participating parent” – line 80

Author response:  Line was altered slightly and now reads “This questionnaire included age, child gender, gender of the participating parent, and family socio economic status. 

Reviewer Comment: Need to capitalize “parents” in line 119

Author response:  We apologize, but I am not sure why parents should be capitalized in this sentence?

Reviewer Comment:  Incomplete sentence – line 132-133

Author response:  This sentence was corrected to read, “Finally, to assess the mediation model, the Structural Equation Modeling (SEM) was implemented.”

Reviewer Comment:  Multiple types on the 2nd sentence of the discussion (lines 190-193)

Author response:  checked